# Spotlight on the Transglutaminase 2-Heparan Sulfate Interaction

**DOI:** 10.3390/medsci7010005

**Published:** 2019-01-04

**Authors:** Giulia Furini, Elisabetta A.M. Verderio

**Affiliations:** 1School of Science and Technology, Nottingham Trent University, Nottingham NG11 8NS, UK; giuliafurini@gmail.com; 2BiGeA, University of Bologna, 40126 Bologna, Italy

**Keywords:** transglutaminase-2 (TG2), heparan sulfate proteoglycans (HSPGs), syndecan-4 (Sdc4), fibrosis

## Abstract

Heparan sulfate proteoglycans (HSPGs), syndecan-4 (Sdc4) especially, have been suggested as potential partners of transglutaminase-2 (TG2) in kidney and cardiac fibrosis, metastatic cancer, neurodegeneration and coeliac disease. The proposed role for HSPGs in the trafficking of TG2 at the cell surface and in the extracellular matrix (ECM) has been linked to the fibrogenic action of TG2 in experimental models of kidney fibrosis. As the TG2-HSPG interaction is largely mediated by the heparan sulfate (HS) chains of proteoglycans, in the past few years a number of studies have investigated the affinity of TG2 for HS, and the TG2 heparin binding site has been mapped with alternative outlooks. In this review, we aim to provide a compendium of the main literature available on the interaction of TG2 with HS, with reference to the pathological processes in which extracellular TG2 plays a role.

## 1. Extracellular Transglutaminase 2 in Human Pathology

Transglutaminase-2 (TG2) is the most ubiquitous isoenzyme of the transglutaminase (TG) family, present in virtually all tissues and cell types and involved in a large spectrum of biological functions [1,2,3]. Similarly to all the catalytically active members of the family, TG2 catalyses a calcium-dependent transamidation reaction between peptide-bound lysine and glutamine residues, with formation of inter or intramolecular ɛ-(γ-glutamyl)lysine bonds. TG2-crosslinked products are highly resistant to degradation and can reach large molecular sizes [4]. Through transamidation, TG2 also promotes calcium-dependent amine incorporation, typically by incorporating polyamines into peptide-bound glutamine residues [1].

Intracellular TG2 is inhibited by guanine and adenine nucleotides and low calcium level (0.1 μmol·L^−1^); increases in intracellular calcium (0.5–1.5 mmol·L^−1^) or the concentration of calcium in the extracellular environment activate TG2 transamidation, as recently reviewed [5]. Substrates of TG2 transamidation have been collected in the TRANSDAB database (http://genomics.dote.hu/wiki/) [6] (up to 2010), but further interactors and numerous substrates have been revealed in recent proteomic studies [7,8,9]. Based on the conformational heparin binding site of TG2 [10], discussed in this review (Section 6), heparan sulfate (HS) elements of proteoglycans may represent a further level of control of TG2 transamidation. The enzyme is also capable of calcium-independent enzymatic activities, such as guanosine triphosphate (GTP) binding and hydrolysis [11,12,13,14], kinase activity [15,16,17] and protein disulphide isomerase activity [18,19,20,21]. Furthermore, TG2 can catalyse protein deamidation, with important implications in celiac disease [22,23,24,25,26].

TG2 was originally described as a cytosolic protein, however, we now know that TG2 is located at the cell surface and extracellular space of many cell types, where it promotes stabilisation and deposition of the extracellular matrix (ECM) via its calcium-dependent crosslinking activity [27,28,29,30]. In the extracellular space, TG2 has a further non-enzymatic role, acting as a structural protein in complex with fibronectin (FN) and interacting with integrins and heparan sulfate proteoglycans (HSPGs) to promote cell adhesion and spreading [30,31,32,33,34,35,36].

The role of extracellular TG2 in both physiological and pathological scarring processes has been well described [37]. In wound healing and tissue fibrosis, TG2 modification of ECM proteins favours ECM deposition, stabilisation and resistance to proteolytic decay, providing a matrix-crosslinked platform for the adhesion and migration of cells such as matrix-secreting fibroblasts or endothelial cells [27,28,38,39,40,41,42,43,44,45,46,47]. The non-enzymatic activity of TG2 as a scaffold protein promotes cell adhesion and migration, especially in the context of matrix fragmentation/cell injury [32,33,34]. TG2 contributes to the uncontrolled matrix deposition underlying pathological conditions such as tissue scarring, kidney fibrosis [37,41,48,49,50,51], liver fibrosis [52,53], heart fibrosis [54] and pulmonary fibrosis [55,56]. Beside direct matrix stabilisation, TG2 transamidating activity has been suggested to activate the pro-fibrotic cytokine transforming growth factor-β (TGF-β) by matrix recruitment, via crosslinking of the latent TGF-β binding protein (LTBP), and release of the active cytokine from its latency binding complex [57,58,59,60]. TGF-β has been reported to be activated in a TG2-dependent manner in animal and cell models of kidney fibrosis, and TG2-knockout results in lower TGF-β activation *in vivo* [9,61,62,63,64].

Beside tissue fibrosis, TG2 activity has been associated with a number of other pathological conditions, such as celiac disease, hepatic disease, arthritis, cardiovascular diseases, atherosclerosis [65], neurodegeneration [66,67,68,69] and cancer [70,71,72,73,74,75,76,77]. Often, the pathological role of TG2 implies its secretion in the extracellular environment.

The mechanism of TG2 secretion has been enigmatic as TG2 lacks a leader peptide (signal peptide) necessary for endoplasmic reticulum (ER) targeting and ER-to-Golgi classical protein secretion [78,79], as well as Golgi-associated protein modifications such as glycosylation [79]. Therefore, TG2 has been suggested to be secreted through a non-classical pathway [30,80]. A number of recent studies have investigated the role of HSPGs in the enzyme’s externalisation. TG2 interacts with HS chains of HSPGs and heparin, a highly sulfated analogue of HS, with affinity comparable to that of TG2 for FN. The interaction of TG2 with HS chains of cell surface syndecans has the potential to shape the pathophysiological role of TG2.

In the next paragraphs, we will describe the main characteristics and biological roles of HSPGs and then comprehensively review the literature that has investigated the interaction between HSPGs and TG2 in the past 15 years.

## 2. The Diversity of Heparan Sulfate Proteoglycans Functions in the Cells

Heparan sulfate proteoglycans are specialised glycoproteins characterised by a core protein and one or more HS chains, long glycosaminoglycan (GAG) carbohydrates (40–300 monosaccharides, covering 20–150 nm), strongly anionic and characterised by different levels of sulfation and epimerization [81,82]. In mammals, nine families of HSPGs are recognised depending on location, core protein and chain features [82]. Based on location, they can be broadly distinguished into two large categories, cell surface membrane-associated proteoglycans and HSPGs secreted in the ECM, mainly located in the basement membrane. The best known cell surface HSPGs are the syndecans and the glypicans, while among the major secreted HSPGs, perlecan, agrin and collagen XVIII-endostatin play a role in the definition of the basement membrane structure, forming a negatively charged protective barrier against the filtration of specific solutes [83,84]. The syndecan family consists of four distinct cell surface HSPGs, syndecan-1–4 (Sdc1–4). Syndecans are single transmembrane receptors consisting of a small cytoplasmic domain, involved in the interaction with cytoskeleton and signal transduction, a transmembrane portion, involved in oligomerisation, and a larger extracellular domain. This comprises from two to five GAG chains, mainly HS, but may include one or two chondroitin sulfate (CS) chains (Sdc1 an Sdc3) [85,86,87,88]. Given their configuration, syndecans provide a link between the actin cytoskeleton and the ECM [88]. Syndecan-4 (Sdc4) (or ryudocan, or amphiglycan), the most studied member of the syndecan family, is ubiquitously expressed in mammalian cells at all stages of development [89].

The HS chains of HSPGs form by a set of enzymatic steps in the Golgi stacks, relying on nucleotide-monosaccharides imported from the cytoplasm [90,91]. The basic elements are disaccharide units of β-d-Glucuronic acid (GlcA) and α-d-*N*-acetylglucosamine (GlcNAc), however, during biosynthesis, a series of enzymes contribute to GlcNAc N-deacetylation/N-sulfation to N-sulfoglucosamine (GlcNS), epimerisation of the GlcA residues to iduronic acid (IdoA) and O-sulfation of IdoA and GlcNS/GlcNAc by specific *O*-sulfotransferases [90,92]. These modifications generate relatively short segments of sulfated sugars (sulfated domains) and iduronic acid alternated to variable lengths of unmodified domains. Once externalised from cells, HSPGs can undergo further modifications, such as selective removal of sulfate groups by plasma membrane-bound endosulfatases (SULF1/2) or specific enzymatic cleavage mediated by heparanase (endo-β-glucuronidase). The latter digests HS between a GlcA-GlcNS pair after an IdoA-GlcNAc dimer, determining the release of chains and associated ligands in the extracellular space [93].

Because of their strong negative charge and the specific modification patterns, the HS chains can bind a large number of ligands, including cytokines, growth factors and ECM structural proteins, thus affecting their activity, stability and localisation in the matrix. Furthermore, HS can act as matrix storage reservoirs for these proteins, providing protection against extracellular proteases and regulating cell responses by inducing ligand oligomerisation and receptor clustering [91,93,94,95,96,97,98]. The long HS chains can cross the ECM, producing both in *cis* (on the same cell) and in *trans* (between neighbour cells) effects [91]. In signal transduction, cell surface syndecans induce a signalling cascade in a non-enzymatic, protein-kinase dependent manner, triggered by ligand binding and specific oligomerisation of the protein core [86,88,99].

Syndecans are involved in cell adhesion and proliferation, both independently or by interacting with integrins [86,88,100,101,102]. By interacting with the heparin-binding domain of FN, they contribute to RGD (arginine-glycine-aspartic acid) independent focal adhesion, inducing stress fibres formation and cell spreading/migration [103,104,105,106,107]. For this reason, Sdc4 is considered a central mediator of cell adhesion, migration and wound response, which in turn upregulate Sdc4 expression [86,108,109,110,111]. In this process, Sdc4 activates protein kinase Cα (PKCα)-dependent signalling cascades, through binding to phosphatidylinositol 4,5-bisphosphate (PIP2) on its cytosolic variable region [112,113,114,115,116,117,118]. However, Sdc4 has been reported to control specific integrin recycling in a PKCα-independent but tyrosine-kinase Src and syntenin-dependent manner, mediating alternation of focal adhesion stabilisation and turnover, and ultimately regulating cell migration [119].

Cell surface HSPGs (both syndecans and glypicans) can act as endocytic receptors, involved in both constitutive and ligand-induced endocytosis in a clathrin, caveolin and dynamin–independent way. This is related to lipid rafts and has been suggested to be similar to micropinocytosis [86,120,121,122,123,124]. In this way, syndecans and associated ligands can be recycled [123] in a process that involves alternate syntenin-PDZ (postsynaptic density protein, disc-large, zonulin-1) domain binding to the cytosolic tail of syndecan and PIP2-syntenin association on the endosome, and requires Arf6 activity [119,125]. Syndecan-4 has also been reported to support caveolin-mediated endocytosis of α5β4 integrins, which promotes fibroblasts and keratinocytes migration and wound closure [126], by mediating PKCα-dependent modulation of Rho GTPase upon HS engagement to FN [126,127,128].

Work from Baietti et al. (2012) has proposed a role for syndecans in the biogenesis and cargo loading of exosomes, in complex with syntenin and Alix, an auxiliary component of endosomal sorting complexes required for transport (ESCRT) machinery, involved in intraluminal vesicles formation [129]. The process would be triggered upon ligand binding to the HS chains of syndecan on the endosome, and syndecan clustering, which would stimulate the recruitment of syntenin-Alix complexes (by syntenin binding on the PDZ domain), supporting the intraluminal budding process required for the formation of exosomes at the multivesicular bodies level [129,130,131]. This syndecan-mediated process would be favoured by heparanase, a HS-specific digesting enzyme [131,132], with stimulation of syndecan clustering at the endosomal level. Therefore, syndecans have been implicated in cargo-targeting to the exosome, however the underlying mechanism is not fully characterised. It remains unclear whether cargo-binding to syndecans only occurs at the cell surface (e.g., through endocytosis of syndecan-cargo complex) or also inside the cell.

## 3. Involvement of Heparan Sulfate in Pathology

Given their widespread role in extracellular ligand accumulation/activity and transmembrane signal transduction, HSPGs have been implicated in the pathogenesis of a disparate number of human diseases. Heparan sulfate proteoglycans, and in particular Sdc4, have been linked to conditions related to wound healing and abnormal chronic repair, from tissue fibrosis to cancer. Defects in wound repair have been reported in Sdc4-knockout mice [133], including abnormalities in cardiac healing after myocardial infarction [134], with defects in fibroblast migration and differentiation into myofibroblast. Syndecan-4 has been involved in ECM contraction, TGF-β signal transduction and ERK response during chronic fibrosis [134,135,136]. As a consequence, several studies have linked syndecan family members with pathological tissue scarring, such as heart disease-associated cardiac fibrosis (e.g., infarction, hypertension) [137,138], pulmonary fibrosis [139] and kidney fibrosis underlying chronic kidney disease (CKD) [140,141,142,143,144,145]. Syndecan-2 (Sdc2) was found to be over-expressed in skin and lung fibrosis [146], although it was recently reported to ameliorate radiation-induced fibrosis in transgenic mice [147]. Syndecan-4 knockout was protective in two experimental models of CKD, the unilateral ureteric obstruction (UUO) and aristolochic acid nephropathy (AAN) models [148]. In a further rat model of CKD established by subtotal nephrectomy (SNx), which well mimics human pathology, Sdc4 was the highest expressed syndecan in fibrotic kidneys, increasing in parallel with the loss of kidney function and peaking at a level of advanced fibrosis, when the process becomes irreversible [63], confirming a clear involvement of Sdc4 in CKD. Furthermore, increased 6-*O* sulfation generated by glucosaminyl-6-*O* sulfotransferase was observed in the UUO model and in human renal allografts, suggesting a role for HS in chronic allograft dysfunction through growth factor-receptor binding regulation [149].

The involvement of HSPGs in tumorigenesis and cancer progression has been investigated thoroughly [150,151]. Cell surface and secreted HSPGs have been shown to take part in most cancer cell features, including cell proliferation, migration, angiogenesis, immune evasion and apoptosis. HS fragmentation and shedding is associated with poor prognosis [151], with the enzyme heparanase being a proposed target for cancer control [152,153]. On the other hand, HSPGs do also play a protective role by inducing cell differentiation [154], promoting natural killer cells response [155,156] or repressing angiogenesis [157,158,159].

HSPGs have also been linked with the pathogenesis of Alzheimer’s and Parkinson’s diseases and with pathological accumulation of protein aggregates, as recently reviewed [160].

## 4. Role of Heparan Sulfate /Syndecan-4 in the Trafficking and Extracellular Function of TG2

Early work from Bergamini’s group introduced heparin, a highly sulfated analogue of the HS chains, in the TG2 precipitation step to improve TG2 purification by affinity chromatography [161]. Verderio and Griffin explored the interaction of TG2 with heparin/HS and the significance of this in mammalian cell systems [32]. It was found that, in situations of matrix fragmentation, with loss of direct cell adhesion of integrin to the RGD sequence of FN, TG2 is able to act as a structural protein, rescuing cell adhesion to FN and averting anoikis through interaction with cell surface HS. This newly uncovered RGD-independent cell adhesion process mediated by TG2 immobilised on FN critically depends on the HS chains of HSPGs, as digestion of HS chains in human osteoblasts completely abolished RGD-independent cell adhesion and spreading onto a TG2-FN matrix [32]. The process depends on the direct binding of HS chains to TG2 and does not require HS chains binding to FN; in fact, saturation of the FN heparin-binding site in mouse fibroblasts did not affect cell adhesion to TG2-FN [33]. In particular, Sdc4 is central in the promotion of RGD-independent cell adhesion via TG2-FN complex, since Sdc4-null fibroblasts were not able to undergo this adhesion process, which was restored by Sdc4-add back to the knockout cells [33]. Cell adhesion to TG2-FN induced a Sdc4-dependent PKCα-signalling cascade as well as focal adhesion kinase (FAK) signalling, mediating the formation of focal adhesions and promoting cell survival in situations of matrix fragmentation [32,33]. Syndecan-4 interaction with TG2-FN complex was suggested to lead to β1 integrins activation via PKCα, with subsequent integrin mediated activation of the FAK downstream signalling cascade in RGD-independent cell adhesion [33]. In support of this observation, PKCα binding mutant Sdc4 was unable to restore cell adhesion and spreading on TG2-FN matrix [33]. Scarpellini et al. (2009) identified HS as the main interactors in Sdc4/HSPGs, as TG2 co-precipitated less with Sdc4 when human osteoblasts, notoriously rich in HS, were treated with bacterial heparitinase, which digests cell surface HS [162].

Griffin and co-workers proposed that a further member of the syndecan family, Sdc2, is involved in RGD-independent cell adhesion to TG2-FN complex. However, Sdc2 did not appear to interact with TG2 directly, and rather it acted as a downstream PKCα-dependent signal transducer regulating actin cytoskeleton in these studies [34,35].

Work from Lortat-Jacob et al. (2012) confirmed the role of HS-binding in cell adhesion to TG2-FN, since recombinant TG2 mutants lacking the proposed heparin binding domain were not conducive of RGD-independent cell adhesion, in contrast to wild-type TG2 bound to FN [10].

The interaction of cell-surface TG2 with heparin/HS was studied by Scarpellini et al. (2009), both in real-time by surface plasmon resonance (SPR), and at equilibrium by solid phase binding assays to immobilised heparin and HS. Purified guinea pig liver TG2 could bind both heparin and HS with high affinity (K_D_ in the low nanomolar range), similarly to TG2 binding to the well-known partner FN [162]. By employing TG2-null mouse epithelial fibroblasts (MEF), it was seen that cellular TG2 was critical to mediate the binding of cells to HS/heparin, as TG2 knockout cells did not adhere to HS/heparin to the same extent as wild type cells [162]. Therefore, this work showed that not only cell surface HS interacts with matrix-bound TG2, but also cellular/cell surface TG2 interacts with matrix HS during cell adhesion [163].

Further investigations explored the possible role for the long chains of HS in the trafficking of TG2 from the cell surface to the matrix. Sdc4-null mouse dermal fibroblasts (MDF) displayed lower cell surface TG2 transamidating activity compared to wild-type MDF, which was restored by Sdc4-add back. Sdc4-knockout cells were characterised by increased cytosolic TG2 and lower membrane-associated TG2 compared to wild type cells, without a change in the total level of TG2 [162]. A HS chemical antagonist (Surfen) [164], or HS digestion by bacterial heparitinase, similarly affected TG2 cell surface activity, raising the hypothesis that Sdc4 is important to “trap” TG2 at the cell surface [162]. A role for heparin/HS in promoting TG2 crosslinking activity was described by Dieker and colleagues (2009) in chondrocytes [165].

Given the role played by HS/Sdc4 in TG2 trafficking to the cell surface [162], the possible underlying mechanism was investigated by our group. A first hypothesis was that HS chains could facilitate TG2 externalisation as observed for fibroblast growth factor 2 (FGF2), a leaderless growth factor [166,167,168,169,170,171]. Fibroblast growth factor 2 is secreted by direct translocation through the plasma membrane in a phosphorylation-dependent manner, involving binding to PIP2 and oligomerisation of the protein with consequent formation of a hydrophilic plasma membrane pore. According to this theory, membrane-proximal HS facilitate FGF2 transport across the pore and towards the outside by means of a trapping mechanism [166,167,168,169,170,171]. We therefore asked whether TG2 could be exported via a similar mechanism involving cell surface Sdc4 HS chains. However, our initial unpublished data do not seem to suggest TG2 oligomerisation or phosphorylation upon stress-induced TG2 externalisation from cells.

In order to shed light on the mechanism, we built the TG2 interactome in the membranes of the UUO kidney. We did so by affinity isolation of TG2 from wild-type and control TG2-knockout kidneys, and identification of the associated proteome by comparative proteomics, which essentially led to the subtraction of all unspecific TG2 partners obtained from the TG2-knockout kidney from the wild type kidney [9]. Transglutaminase-2 was largely associated with proteins typically present in extracellular vesicles of endosomal origin, including the exosome markers Alix, tumor susceptibility gene 101 protein (TSG101) and flotillin among others [9]. Moreover, as previously mentioned, Sdc4 was a specific partner of TG2 in the UUO phenotype. The presence of TG2 in extracellular vesicles of the exosomal type was reported before in other systems [172,173,174], suggesting the requirement of membrane fusion events for TG2 to be secreted [175,176,177]. As both Sdc4 and Alix are known to cooperate in exosome biogenesis [129,131], and were both interacting partners of TG2 in the UUO kidney, where TG2 is largely externalised [9], we hypothesised that TG2 could be released from kidney tubular epithelial cells via membrane vesicles of endosomal origin (such as exosomes). Analysis of the secretome of NRK-52 renal epithelial cells transfected with EGFP-TG2 revealed that TG2 is present in exosomes, and further enriched in exosomes upon TGF-β treatment [9]. Transglutaminase-2 was also found to weakly associate with ectosomes, extracellular vesicles directly shed from the plasma membrane, but it was not present (if not in low amount) as free in the conditioned medium. Likewise, TG2 was detected in the urinary exosomes of CKD patients by Western blotting of extracellular vesicles’ lysates, but not in the vesicle-free urine [9]. Knockout of Sdc4 by short interfering RNA (siRNA) resulted in a significant reduction of TG2 in exosomes, but not in a concomitant decrease of the vesicular protein marker flotillin [9], suggesting that TG2 is recruited to exosomes by Sdc4. This could occur via a direct event, for example, via HS binding, although an indirect event through other shared partners of TG2 and Sdc4 has not been ruled out. Although the association of TG2 with Sdc4 in endosomes has not been investigated, the finding that TG2 co-precipitates with Sdc4 in exosomes implies their interaction at this level [9]. Furthermore, our initial data suggest that TG2 is capable of calcium-dependent transamidation on the surface of the exosomes in the culture medium, as evidenced by a sensitive TG activity assay [9], hence it is able to fulfill the well-known extracellular functions. Table 1 lists the main studies which have reported on the interaction of TG2 with HS/Sdc4, and Figure 1 visually shows the possible interplay of TG2 with HS/Sdc4.

## 5. Partnership of Transglutaminase-2 and Heparan Sulfate/Syndecan-4 in Disease

Despite the potential implications of the TG2-HSPG interaction in both the regulation of TG2 (trafficking, activity) and the role of HS, the relevance of this interaction in human pathology has just begun to be explored.

Tesaluu et al. (2012) have shown that anti-TG2 IgAs-rich auto-antibodies isolated from coeliac disease (CD) patients’ serum were able to reduce the binding of TG2 to HS/heparin, and that the CD autoantibodies affected the adhesion of intestinal epithelial Caco-2 cells to FN-TG2 coated wells [179,180]. They concluded that CD autoantibodies could influence cell adhesion by interfering with HS-TG2 interaction with possible implications in CD pathogenesis.

TG2 has been involved in the process of removal of unwanted cells by macrophages during apoptosis, and TG2^−/−^ mice have been reported to have defective clearance of apoptotic cells and reduced TGF-β1 [186]. Work by Nadella et al. (2015) has shown that, in differentiated macrophages, the interaction with Sdc4-HS enhanced the extracellular activity of TG2, which was critical to macrophages migration towards apoptotic cells and apoptotic cell clearance [183]. Therefore, the TG2-HS partnership appears to affect key cell functions.

In some cases, TG2 can compete with other heparin binding factors or cytokines for HS binding, thus acting as an inhibitor of HS function. An example was provided by Beckouche et al. (2015), reporting that TG2 interfered with the interaction between the HS chains and vascular endothelial growth factor (VEGF), leading to disruption of the VEGF receptor–2 (VEGFR2) signalling pathway and inhibition of angiogenesis, *ex vivo* in retina cells and *in vitro* in endothelial cells [178].

Because of the fibrogenic role played by extracellular TG2 during the progression of renal fibrosis, the effect of Sdc4 in TG2 trafficking and in the development of kidney fibrosis has been extensively investigated both *in vivo*, using murine CKD models, and *ex vivo* in tubular epithelial cells [9,63,148]. In a first study, Sdc4-null mice were subjected to two different experimental models of CKD, the UUO and the AAN. In both models, knockout of Sdc4, which was protective against the progression of CKD, led to lower accumulation of TG2 antigen and activity in the tubulointerstitium [148]. Live imaging of primary kidney tubular epithelial cells isolated from Sdc4-null and wild type mice [9] suggested that Sdc4 knockout alters the cell surface trafficking of TG2. Therefore, Sdc4 and TG2 cooperate in the progression of kidney fibrosis in CKD. This was reinforced by gene expression analysis of transglutaminase and syndecan family members in the SNx rat model of kidney fibrosis [63]. A significant correlation between Sdc4 and TG2 expression during the progression of the disease was established [63]. Furthermore, TG2 was found to co-localise with Sdc4, mainly with HS chains, in cryosections of fibrotic kidney tissue. The co-localisation was predominant in the extracellular interstitial space and peritubular area, and was abolished by loss of HS chains, showing the requirement for HSPGs/Sdc4 in the extracellular localisation of TG2 [63]. In support of a mechanistic role of the TG2-HS interaction in fibrosis, tubular kidney cells pre-treated with the HS antagonist Surfen displayed a lower level of active TGF-β over total TGF-β, which was accompanied by lower matrix recruitment of extracellular TG2, when the percentage of active TGF-β and/or the TGF-β-downstream transcriptional signalling (Smad3 phosphorylation) was measured [63,148]. Therefore, the binding of extracellular TG2 to HS appears to affect the activation of TGF-β, which may be due to increased matrix crosslinking hence retention of LTBP [63]. On the other hand, extracellular TG2 produces matrix stabilisation by forming ε(γ-glutamyl)lysine crosslinks, and another possible theory is that increased matrix stiffening by TG2, jointly with latency associated peptide (LAP) binding to cell surface integrin, could produce the mechanical tension necessary for TGF-β activation [63,187,188,189]. In this scenario, lack of HS/Sdc4 could result into lower TG2 externalisation and, as a consequence, less TGF-β activation. Once exported, TG2 could probably interact with the HS chains of other extracellular HSPGs, such as the basement membrane perlecan, which are likely to contribute to the enzyme-trafficking in the extracellular space and localisation in the matrix. The specific presence of both Sdc4 and perlecan in the membrane interactome of TG2, recently reported in the murine UUO model of CKD, cemented the idea of a strong partnership of TG2 with HSPG during fibrosis progression [9]. The group of Griffin developed a selective peptidomimetic inhibitor of TG2 which impeded the binding of TG2 to Sdc4 and also affected TG2 translocation into the extracellular matrix; this inhibitor led to lower fibronectin deposition in NIH-3T3 cells [184,190] and considerable reduction in collagen deposition in a mouse model of hypertensive nephrosclerosis [184,190]. Further work showed that the TG2 selective inhibitor 1–155 reduced the development of cardiac fibrosis both *in vitro* and in two experimental mouse models of cardiac fibrosis. The TG2 inhibitor blocked TG2 expression/externalisation in TGF-β1-treated c ardiofibroblasts by affecting the TG2-Sdc4 interaction [54]. Therefore, the interplay between TG2 and HS appears to have wide significance in the fibrotic process of a variety of organs and tissues.

Few studies have begun to explore the importance of the TG2-HS interaction in cancer progression. Sdc4 and TG2 upregulation have been involved in metastasis and tumor survival in renal carcinomas, potentially via cell signaling involving integrin β1 [191]. Similarly, upregulation of Sdc4 was observed in the highly metastatic cell line KP1, and the cell surface HS chains were proven important for TG2/S100A4 mediated cell spreading in this cell line, by binding TG2-crosslinked protein S100A4 and promoting PKC-α signal transduction [185]. Dieker and colleagues also suggested a role for HS chains in TG2 crosslinking-mediated oligomerization of sonic hedgehog (Shh) protein, resulting in an enhanced signaling involved in tumor progression [165].

## 6. The Heparin Binding Site of Transglutaminase-2

The ability of proteins to bind HS is likely to depend on the presence of narrow pockets of positively charged basic amino acids (lysine, arginine and rarely histidine) on their surface in the tertiary folded structure [82,93]. Two consensus motifs have been proposed: XBBXBX and XBBBXXBX, where B represents a basic residue and X either a hydropathic—neutral or hydrophobic residue [192]. However, these are not necessarily typical of all heparin binding proteins [93]. In general, the specificity of the interactions of HS chains with proteins is associated with the overall organization of the HS chain, such as the presence of sulfated IdoA and the number of 6*O* sulfation, and not with a specific monosaccharide sequence [193].

Three different research groups have described the heparin binding site of TG2. Results were published in the same year, but there was no definitive agreement on the conclusions [10,180,181] (Figure 2, Table 2).

Teesalu and colleagues [180] investigated the heparin binding sites of TG2 using synthetic TG2 peptides. With a surface plasmon resonance (SPR)-reliant approach, the group investigated the heparin affinity of five different TG2 peptides, 11-14 amino acids long (P1: 202-KFLKNAGRDCSRRS-215; P2: 261-LRRWKNHGCQRVKY-274; P3: 476-RIRVGQSMNMGS-487; P4: 590-KIRILGEPKQKRKL-603; P5: 671-DKLKAVKGFRN-681). Peptide P1 and peptide P2 showed the highest heparin affinity by SPR. They also displayed the highest immunoreactivity towards IgA anti-TG2 autoantibodies obtained from patients with CD [180], which had been suggested to interfere with TG2-HS binding [179]. Conversely, peptide P4 showed only a minimal association with heparin by SPR analysis [180]. The heparin-binding TG2 peptides P1 and P2 are closely located on the surface of the catalytic domain, as part of α-helical structures [180]; moreover, peptide P2 contains the typical consensus sequence XBBXBX for heparin binding (261LRRWKN266) [192]. Peptide P2 was able to significantly interfere with RGD-independent cell adhesion to TG2-FN heterocomplex [180], which, as reported before, is controlled by direct interaction of matrix-bound TG2 with cell surface HS [32]. Therefore, TG2 peptide 261-LRRWKNHGCQRVKY-274 (P2) emerged as a heparin binding peptide in this study.

The Griffin group [181] proposed that two peptide sequences of TG2, 590-KIRILGEPKQKRK-602 (HS1), located at the tip of C-terminal β barrel 2, and 202-KFLKNAGRDCSRRSSPVYVGR-222 plus K-387 (HS2), forming a narrow pocket lined with basic residues in the three-dimensional structure, could serve as heparin-binding sites allowing simultaneous binding of FN. To investigate the heparin binding properties of these two putative heparin-binding sequences, the group produced mammalian transfection plasmids expressing TG2 with mutations in key basic residues, resulting in mutant HS1 (K600A, R601A, K602A) and mutant HS2 (K205A, R209A) [181]. The constructs were transfected into cells and the total cell lysates analysed for heparin binding using a heparin-sepharose purification column. Overexpression of the TG2 mutants into HEK293/T17 cells led to reduced TG2 binding of the cell lysates to the heparin-sepharose purification column compared to cells transfected with wild-type TG2 [181]. Since HS2 TG2 mutants showed the lowest binding, this suggested that the basic residues of this sequence (K205 and R209) are potentially critical for heparin/HS association with TG2.

Upon *in silico* structural analysis, the two clusters docked well with heparin/HS when TG2 was in its closed conformation [181]. Consistent with this, high affinity of TG2-expressing cell lysate for the heparin-sepharose column was observed in the presence of GTP, which favours the closed conformation of TG2 [194,195], while employment of the active site inhibitor R281 or cell transfection with an active site cysteine-277 TG2 mutant, which blocks the enzyme in an open conformation, reduced the affinity of the cell lysate for heparin-sepharose [181]. A synthetic peptide corresponding to the HS2 region from position 200 to position 216 of wild-type TG2: 200-NPKFLKNAGRDCSRRSS-216 (peptide P1) [181], was tested in a solid binding assay versus human recombinant Sdc4 and Sdc2, using a scrambled peptide as a control [181]. Peptide P1 strongly bound Sdc4, while binding to Sdc2 was almost null, suggesting that the 200–216 TG2 region binds preferentially Sdc4. The same peptide partially interfered with endogenous TG2 co-precipitation with Sdc4 from cell lysate of Swiss 3T3 cells overexpressing TG2 with tetracycline-inducible system [40], and was able to support RGD-independent cell adhesion to FN via activation of a PKC-α signalling cascade [181]. The heparin binding site mimicking peptide was able to interfere with TG2-Sdc4 binding in a dose-dependent manner, affecting macrophages migration to apoptotic cells and apoptotic cell clearance [183]. Recently, new TG2 inhibitors have been reported to interfere with the TG2-Sdc4 complex, TG2 translocation and extracellular activity via blocking TG2 in the open conformation. Furthermore, the inhibition slowed down cell migration, attenuated angiotensinogen II-induced kidney fibrosis and myocardial infarction-induced loss of cardiac function/ fibrosis [54,184]. Therefore, TG2 peptide 200-NPKFLKNAGRDCSRRSS-216 emerged as a heparin binding sequence in this study.

The Verderio group identified possible heparin binding consensus motifs in the TG2 sequence and produced recombinant human TG2 mutants targeting these domains by site-directed mutagenesis. As a result, nine recombinant mutant TG2 proteins were expressed and purified: M1a (R262S), M1b (R263S) and M1c (K265S), independently targeting 262-RRWK-265 cluster; M2 (K202S/K205S); M3 (K598S/K600S/R601S/K602S), targeting the 598-KQKRK-602 cluster; M4 (R19S); M5 (R28S); M6 (R580S); M7 (K634S) [10]. The heparin binding properties of these mutants were investigated by SPR on immobilised heparin (biotinylated heparin bound to the streptavidin surface of a Biacore sensorchip). All three M1 mutants affecting the 262-RRWK-265 cluster had a strong decrease in affinity for heparin compared to the wild-type recombinant protein. The M3 mutant, affecting the 598-KQKRK-602 cluster, also displayed an almost complete loss of heparin binding [10]. Mutations in single residues R19 (M4) and R28 (M5) caused a significant reduction in affinity for heparin, and in K634 (M7) led to a change in heparin recognition, which supported the role of these residues in heparin-binding [10]. TG2 mutants M1c and M3, which had virtually no residual heparin affinity, also failed to support RGD-independent cell adhesion [10]. From these results, cluster 262-RRWK-265 and 598-KQKRK-602 (with K600 being the most important residue of the cluster) were suggested as the main elements of the heparin binding site of TG2, both crucial for heparin binding, with the possible participation of the basic residues R19, R28 and K634 [10]. Interestingly, the main clusters 262-RRWK-265 and 598-KQKRK-602 are close to each other on the surface of TG2 when the enzyme is in the folded conformation, forming a composite binding domain lined with basic residues [10]. Examination of the three-dimensional protein structure revealed that basic residues in close proximity to these two clusters, R19 and R28 and K634, could be involved in heparin interaction [10]. Molecular modeling revealed that this heparin binding site can make contact with a single heparin-derived pentasaccharide [10]. Multiple alignment analysis of these sequences in different taxa and for the different TG family members showed that they are well conserved and typical of the TG2 isoforms only [10]. The two main clusters 262-RRWK-265 and 598-KQKRK-602, which are spatially closed in the folded conformation, are very distant in the open “active” conformation of TG2. Proof that the closed conformation of TG2 was required for the formation of the heparin binding site was obtained by using the commercially available open form of TG2 which displayed significantly less heparin binding compared to TG2 in the presence of the calcium chelator ethylenediaminetetraacetic acid (EDTA) when measured by SPR [10].

An independent study by Beckouche et al., (2015), which relied on a system of TG2-null cells transfected with TG2 M1 mutant cDNA, confirmed that this mutant lacked HS/Sdc4 binding. The TG2 mutant M1 did not interfere with the interaction between HS chains of HSPGs and VEGF, hence did not inhibit angiogenesis compared to wild-type TG2 [178]. Furthermore, TG2 mutants M1c and M3 were less exposed on the surface of cells when transfected into rat renal epithelial cells (unpublished data), confirming the importance of these sites for TG2 cell surface retention and extracellular trafficking via HS binding.

The heparin binding domains collectively proposed by the three groups are summarised in Figure 2, and the different approaches used for the mapping of the TG2 heparin binding site(s) are shown in Table 2. A TG2 region between amino acids 200 and 222 was investigated by each group as potentially critical for heparin binding [10,180,181]. Teesalu et al. pointed at 202-KFLKNAGRDCSRRS-215 and Wang et al. suggested 202-KFLKNAGRDCSRRSSPVYVGR-222 as a putative binding region (Figure 2) [180,181]. Wang and colleagues found K205 and R209 as critical residues for heparin binding [181]. Lortat-Jacob et al. analysed two residues belonging to this region, K202 and K205 (mutant M2), however, these residues were not found critical for heparin binding in this study, and R209 was not specifically investigated in the Verderio group’s study [10,180,181]. Lortat-Jacob et al. identified a first cluster, the 261-LRRWK-265 sequence as central for heparin binding [10], and a second essential cluster, the 598-KQKRK-602 sequence, which overlaps with regions that were also studied by Teesalu et al. (peptide P4, 590KIRILGEPKQKRKL603) and Wang et al. (mutant HS1, 590KIRILGEPKQKRK602, mutations: K600A, R601A, K602A). However, a peptide in this region showed only a weak affinity to heparin by SPR, according to Teesalu et al. [180], and lysates of cells transfected with this mutant displayed no sensible differences in binding to a heparin-sepharose column, according to Wang et al. [181].

Therefore, the three groups are not in agreement on the heparin binding site of TG2 and the three different approaches used may be responsible for this lack of cohesion. Teesalu and colleagues [180] and Lortat-Jacob et al [10] employed SPR to investigate binding affinity of putative TG2 sequences. However, Teesalu et al. made use of synthetic peptides representing the investigated TG2 sequences, while Lortat-Jacob et al. produced recombinant mutant TG2 proteins, which were obtained via site-directed mutagenesis. Wang et al. [181] relied on mammalian HEK293/T17 cells transfected with wild-type and mutant human TG2 cDNAs, the lysates of which were assessed for heparin binding using a heparin-sepharose affinity isolation column.

In a recent study from the Verderio group, the binding proprieties of other members of the transglutaminase family to heparin were tested by SPR [63]. A strong binding of recombinant TG1 for heparin was detected, which was higher than TG2 affinity at the same concentration. However, this strong binding was not validated by an alternative heparin-binding assay, and multimerisation of TG1, which could lead to a false positive result by SPR, was not completely ruled out. TG3 and FXIIIa had only a weak affinity for heparin. The TG2 heparin binding site proposed by the Verderio group (262–RRWK-265, 598–KQKRK-602) [10] has no similar sequences in TG1 and only the basic residue Arg19 is conserved and exposed on the surface of TG1 [63]. The binding site proposed by other groups, such as that at residues 202–215 [181], is also not conserved in TG1. These data suggest that TG1 interaction with heparin/HS may occur through a different binding site [63]. On the other hand, TG3 displays two positive clusters exposed on the surface similarly to TG2, but these could form only a weak heparin binding site (259–KNWK.262 and 606–RVRK-609), and FXIIIa sequence shows no similarities with the TG2 heparin binding site [63]. In conclusion, from the available data, TG2 seems to have a unique heparin binding domain within the TG family.

We previously identified two TG2 variants, along with the canonical TG2, in the rat SNx experimental model of chronic kidney disease and reported that these increase in expression post-SNx [63]. Although the TG2 variants represent a small fraction of the total TG2 transcripts in the rat SNx model, their over-expression during fibrosis progression may escape normal regulatory pathways. In fact, these variants lack the C-terminal GTP-binding site, which in turn inhibits TG2 transamidation, and therefore they might be difficult to control. Interestingly, the C-terminal peptide encoded by the truncated variant TGM2_v2 is a new epitope formed by intron retention [196]. This peptide lacks the crucial basic amino acid cluster at position 598–602, hence the TGM2_v2 variant is predicted not to form the conformational binding site reported for the canonical full length TG2 form (TGM2_v1) [63,196]. Moreover, both TGM2_v2 and TGM2_v4 putative proteins would lack K633 (which corresponds to K634 in human TG2), another important residue for the affinity of TG2 to heparin. Therefore, the HS-TG2 interaction is anticipated to be lost in the truncated variants of TG2. This observation suggests that in fibrotic kidney the TG2 variants would not only escape the normal interaction with GTP but also binding by HS.

## 7. Conclusions

The affinity of TG2 for the HS chains of HSPGs, Sdc4 in particular, has only started to be characterised, as not much is known on the nature of the HS structure and the degree of sulfation required for TG2 binding. We know that Sdc4 plays an emerging role in the cell surface trafficking and extracellular activity of TG2, being implicated in the externalisation pathway of TG2 either by trapping TG2 at the cell surface or by TG2 co-association in extracellular vesicles of endosomal origin. Other HSPGs have been detected in the TG2 interactome, such as perlecan and endostatin, which could modulate TG2 in a similar way, either through direct or indirect interaction. Although definite agreement on the TG2 heparin binding site has not been reached, most groups agree that TG2 binds heparin/HS when in the folded conformation, raising the possibility that HSPGs could not only favour TG2 externalisation but also contribute to modulate the catalytic activity of TG2 acting at the conformation level. Clearly, the affinity of TG2 for HS has opened a new chapter in TG2 biology, which has just begun to be uncovered.

## Figures and Tables

**Figure 1 medsci-07-00005-f001:**
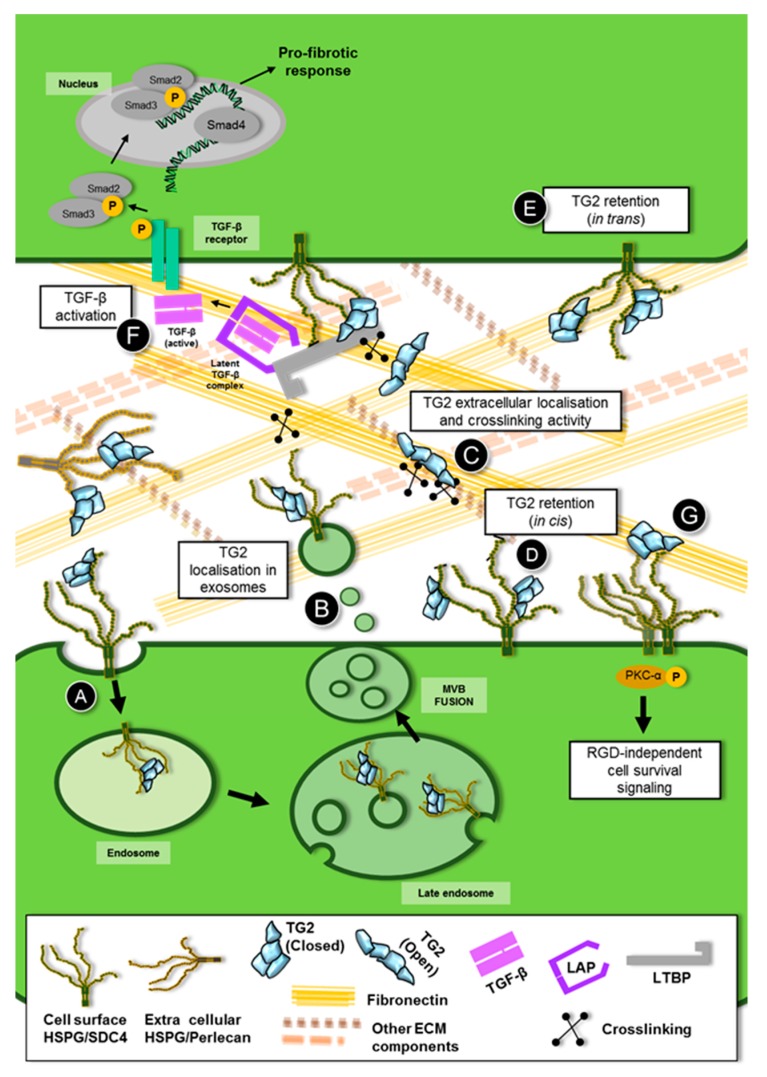
Interplay between transglutaminase-2 (TG2) and heparan sulfate (HS)/syndecan-4 (Sdc4). (**A**) HS/Sdc4 traps TG2 at the cell surface contributing to its extracellular accumulation. This interaction could determine endocytosis of TG2 and incorporation of TG2-HS/Sdc4 complexes in endosomes (**B**) Upon fusion of the outer membrane of multivesicular bodies (MVB) with the plasma membrane, the TG2-bearing exosomes are thought to accumulate in the extracellular matrix (ECM) [9]. (**C**) Once in the extracellular space, TG2 would undergo a conformational change adopting the open conformation due to high calcium/guanosine triphosphate (GTP) ratio, with a lowering or loss of HS binding; the free TG2 could interact with extracellular proteoglycans, ECM fibronectin and other protein partners and switch between the open and the folded HS-bound conformation in a dynamic way. (**D**) Given the length and flexibility of the HS chains, cell surface HS could recruit TG2 not only in *cis* (on the same cell) but also (**E**) in *trans* (from neighbouring cells). (**F**) In turns, extracellular TG2 recruits latent transforming growth factor- β1 (TGF-β1) by matrix crosslinking and cooperates with HS/Sdc4 in the activation of latent TGF-β complex. (**G**) Bound to fibronectin, extracellular TG2 can act as an adhesive protein promoting arginine-glycine-aspartic acid (RGD)-independent cell adhesion via HS/Sdc4, leading to activation of protein kinase Cα (PKCα) and focal adhesion kinase (FAK).

**Figure 2 medsci-07-00005-f002:**
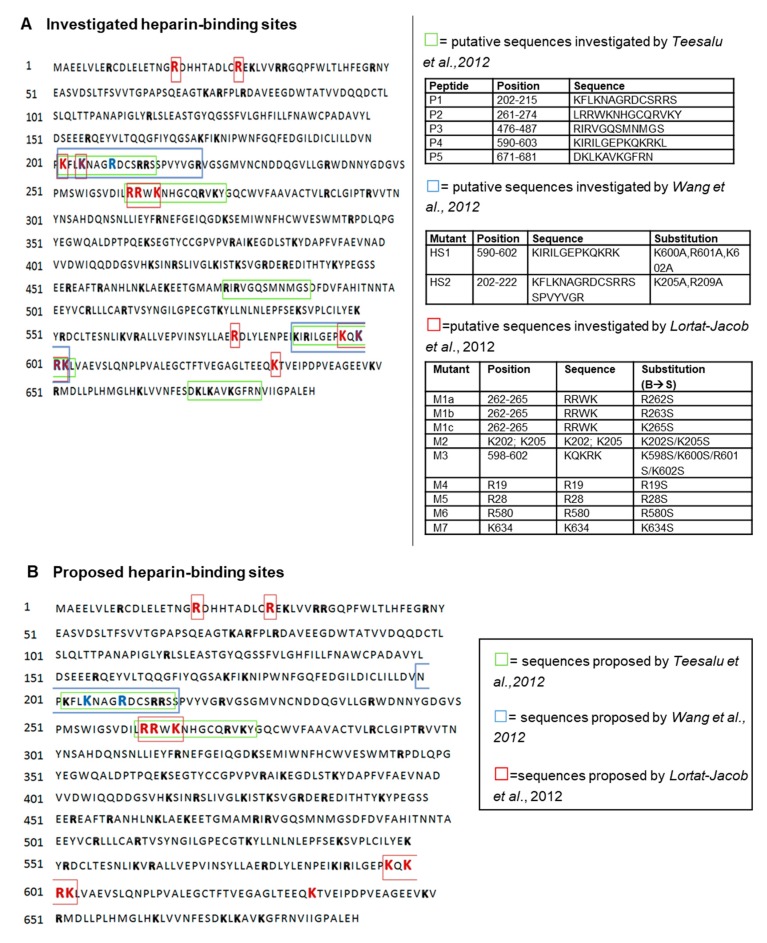
Investigation of the heparin binding site(s) of TG2. (**A**) Putative heparin binding sites independently investigated by three research groups [10,180,181]. (**B**) Heparin binding regions mapped by the three groups [10,180,181]. Basic residues arginine (R) and lysine (K) are highlighted in bold; the amino acids mutated by site directed mutagenesis in the cited studies are highlighted in blue (for Wang et al., 2012) and/or red (for Lortat-Jacob et al., 2012).

**Table 1 medsci-07-00005-t001:** Interactions between transglutaminase-2 and heparan sulfate proteoglycans. TG2, transglutaminase-2; HS, heparan sulfate; HSPGs, heparan sulfate proteoglycans; Sdc4, syndecan-4; FN, fibronectin.

Evidence of TG2-HSPGs Interaction	References
**TG2-HS binding studies**	[10,63,161,162,178,179,180,181,182]
**Mapping of TG2-heparin binding site**	[10,180,181]
**Co-precipitation of TG2 and Sdc4**	[9,33,34,35,63,162,183,184]
**Co-localisation of TG2 and HS/Sdc4**	[63,148,162]
**Interaction between HS and TG2-FN heterocomplex**	[10,32,33,34,35,180,181,185]
**Studies of TG2 in Sdc4 knockout models**	[9,33,148,162]

**Table 2 medsci-07-00005-t002:** Different approaches used for mapping the transglutaminase-2 (TG2) heparin binding site(s). Key amino acids investigated by site directed mutagenesis [10,181] are underlined.

Approach	Proposed Heparin Binding Site(s) of TG2	Reference
Surface plasmon resonance (SPR) to test heparin-affinity of TG2 peptides	LRRWKNHGCQRVKY 261-274 (peptide P2) KFLKNAGRDCSRRS 202-215 (peptide P1)	[180]
Heparin sepharose column to test the affinity of cell lysates of HEK293/T17 cells transfected with human TG2 mutant cDNAs	NPKFLKNAGRDCSRRSS 200-216 (peptide P1)	[181]
Surface plasmon resonance (SPR) to test heparin-affinity of recombinant human TG2 mutants	RRWK 262-265 (mutant M1) KQKRK 598-602 (mutant M3) R19 (mutant M4), R28 (mutant M5) and K634 (mutant M7)	[10]

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
