# Peer review of "Spotlight on the Transglutaminase 2-Heparan Sulfate Interaction"

_medsci, 2019, doi:10.3390/medsci7010005_

Round 1

Reviewer 1 Report

In the present review Furini and Verderio examined the role of transglutaminase 2 (TG2) and heparan sulphate (HS) and their interaction in several human diseases.

Authors discussed mainly about biological interactions between these two molecules. In the paragraph 1, they mentioned that these molecules are involved in the pathogenesis of several disorders, but they did not detail such aspects. Indeed, in its current form, the review is mainly devoted to a basic science public. It is necessary therefore to elucidate how TG2 and HS  are involved in the pathogenesis of several fibrotic diseases (for example kidney fibrosis, liver cirrhosis, atherosclerosis, scleroderma and Crohn’s disease).

Another aspect that should be discussed is the role played by TG2 and HS in cancer. Indeed it is possible that such molecules could promote angiogenesis, tumor growth and vascular spreading, thus leading to metastasis. Additionally, the prognostic value of TG2 and HS should be summarized. Another aspect that should be covered is the role in the interaction with cancer microenvironment and extracellular matrix components, since it may be crucial for tumor spreading.

Author Response

Manuscript ID: medsci-395291

Reviewer 1                

Thank you for assessing our manuscript ‘Spotlight on the transglutaminase 2-heparan sulfate interaction’, and for suggesting how to improve it. Please find our detailed response below.

In the present review Furini and Verderio examined the role of transglutaminase 2 (TG2) and heparan sulphate (HS) and their interaction in several human diseases.

Authors discussed mainly about biological interactions between these two molecules. In the paragraph 1, they mentioned that these molecules are involved in the pathogenesis of several disorders, but they did not detail such aspects. Indeed, in its current form, the review is mainly devoted to a basic science public. It is necessary therefore to elucidate how TG2 and HS  are involved in the pathogenesis of several fibrotic diseases (for example kidney fibrosis, liver cirrhosis, atherosclerosis, scleroderma and Crohn’s disease).

To meet this request we have dedicated a separate section to the role of TG2 and heparan sulfate (HS) in human pathology (new section 5 titled “Partnership of TG2 and HS/Sdc4 in disease”).  Lines 195-199 and 220-255 of the old manuscript have been merged in this new section which touches on the significance of TG2-HS in a variety of conditions from celiac disease to apoptotic cell death. Large emphasis has been given to the importance of the 2 molecules and their interaction in tissue fibrosis from cardiac to kidney fibrosis.

Another aspect that should be discussed is the role played by TG2 and HS in cancer. Indeed it is possible that such molecules could promote angiogenesis, tumor growth and vascular spreading, thus leading to metastasis. Additionally, the prognostic value of TG2 and HS should be summarized. Another aspect that should be covered is the role in the interaction with cancer microenvironment and extracellular matrix components, since it may be crucial for tumor spreading.

Although there is not a vast literature on the topic of TG2-HS interaction in cancer, we tried to review the existing literature from this angle and added a paragraph towards the end of section 5.  There are no indications in the literature of a joint prognostic value of TG2 and HS, as far as we know. As we wanted to specifically review the interaction of TG2 and HS,  we felt that to review the prognostic value of TG2 and HS in general was outside our scope. However, to increase emphasis on disease we have also modified section 2 of the old manuscript (titled “The diversity of heparan sulfate proteoglycans functions in the cells”) to separate fundamental science on heparan sulfate (HS), from the significance of HS in disease. New section 3 on “Involvement of HS in pathology” has been largely re-written to give an improved evidenced-based focus on HS in disease including touching its prognostic value in cancer. 

Thank you for your help in processing and evaluating our work.

Reviewer 2 Report

There are a few previous reviews published that cover some of the ground presented here [Kanchan et al 2015; Verderio and Scarpelimi 2010] but this review is up to date and pushes the subject forward in a significant way and so I think is appropriate for publication at this time. It is quite a mechanistic review and it would be easy, but probably inappropriate, to suggest more depth of consideration of specific disease states that are mentioned (but inevitably not considered in great depth). Indeed, a comparative review in different tissues  would possibly merit a review in itself.  There are a few minor typos and style artifacts (I would not necessarily say errors) but these are easy to correct and I'll itemise these in the comments for editors.

I felt that some sections were a little long between subheadings and there might be room for a few more subheadings  - to help the new reader explore the complex narrative that links two very different systems. For example,  I wanted to check the effect of calcium (needed for Tg2 cross-linking activity) and something that might interact with HS/GAGs and found it difficult to locate. It is a big ask to brig these two very different areas together and it has been done well on the whole - but it is a densely written piece and I'd like to see features of TG2 described in lines 30-38 easier to identify with regard to HS in subsequent text.

On a completely different note considering content I did wonder whether phosphorylation of TG2 at Ser216 might affect binding of heparin binding sites (it appears to be at the end of predicted interactive sequences on TG2) in Table 2 in different cell systems and whether there is any information regarding differential binding of TG2 isoforms to heparin sulphate(isoforms not really mentioned). 

On balance though it is pretty comprehensive and scrolling through recent and key publications on pubmed I was pleased to see that most, if not all obvious references were covered. 

Author Response

Manuscript ID: medsci-395291

Reviewer 2                

Thank you for assessing our manuscript ‘Spotlight on the transglutaminase 2-heparan sulfate interaction’, and for suggesting how to improve it. Please find our detailed response below.

There are a few previous reviews published that cover some of the ground presented here [Kanchan et al 2015; Verderio and Scarpelimi 2010] but this review is up to date and pushes the subject forward in a significant way and so I think is appropriate for publication at this time. It is quite a mechanistic review and it would be easy, but probably inappropriate, to suggest more depth of consideration of specific disease states that are mentioned (but inevitably not considered in great depth). Indeed, a comparative review in different tissues  would possibly merit a review in itself.  There are a few minor typos and style artifacts (I would not necessarily say errors) but these are easy to correct and I'll itemise these in the comments for editors.

Thank you for your appreciation and your time to itemise typos and linguistic weaknesses.

I felt that some sections were a little long between subheadings and there might be room for a few more subheadings  - to help the new reader explore the complex narrative that links two very different systems. For example,  I wanted to check the effect of calcium (needed for Tg2 cross-linking activity) and something that might interact with HS/GAGs and found it difficult to locate. It is a big ask to brig these two very different areas together and it has been done well on the whole - but it is a densely written piece and I'd like to see features of TG2 described in lines 30-38 easier to identify with regard to HS in subsequent text.

At line 30 (new manuscript) we have better specified the effect of calcium and changed the order of some of the facts to ease the reading. Lines 37-40 (new manuscript), on calcium-independent enzymatic activities, are now mentioned after the description of TRANSDAB substrate database, as this is linked to the effect of calcium.

We have also established an earlier link between HS and TG2 activity (expanded later on section 6) at line 35 of new manuscript.

We have broken section 2 of the old manuscript (titled “The diversity of heparan sulfate proteoglycans functions in the cells”) into two subheaded sections, by creating a section 3 titled “Involvement of HS in pathology”, which better focuses on the significance of HS in disease and allows to appreciate the similarities of roles between TG2 and HSPG, in tissue fibrosis.

To better structure the narrative (and also to respond to the request of reviewer 1) we have dedicated a separate section to the role of TG2 and HS in human pathology (new section 5 titled “Partnership of TG2 and HS/Sdc4 in disease”).  

On a completely different note considering content I did wonder whether phosphorylation of TG2 at Ser216 might affect binding of heparin binding sites (it appears to be at the end of predicted interactive sequences on TG2) in Table 2 in different cell systems and whether there is any information regarding differential binding of TG2 isoforms to heparin sulphate(isoforms not really mentioned).

We thank the reviewer for this request. We previously identified two TG2 variants, along with the canonical TG2, in the rat SNX experimental model of chronic kidney disease (Ref Burhan et al, 2016) and reported that these increase in expression post-SNx.  Although the TG2 variants represent a small fraction of the total TG2 transcripts in the rat SNx model, their over-expression during fibrosis progression may escape normal regulatory pathways as these lack the C-terminal GTP-biding site, which inhibits TG2 transamidation; therefore they might be difficult to control.

Furthermore, the C-terminal peptide encoded by TGM2_v2, is a new epitope formed by intron retention. This peptide lacks the crucial basic amino acid cluster at position 598–602, hence the TGM2_v2 variant is predicted not to form the conformational binding site reported for the canonical long TG2 form (TGMs_v1). Moreover, both TGM2_v2 and TGM2_v4 putative proteins would lack K633 (which corresponds to K634 in human TG2), another important residue for the affinity of TG2 to heparin. Therefore the HS-TG2 interaction is anticipated to be lost in the truncated variants of TG2.

This observation suggests that in fibrotic kidney the TG2 variants would not only escape the normal interaction with GTP but also with HS. We have now integrated this part in lines 493-506 (section 6).

We do not know whether phosphorylation of TG2 at Ser216 would affect binding of TG2 to heparin. This is an interesting observation that we hope to pursue in future work.

On balance though it is pretty comprehensive and scrolling through recent and key publications on pubmed I was pleased to see that most, if not all obvious references were covered.

Thank you for your help in processing and evaluating our work.

Round 2

Reviewer 1 Report

The paper can be accepted